# Nanocomposite Electrocatalysts for Hydrogen Evolution Reactions (HERs) for Sustainable and Efficient Hydrogen Energy—Future Prospects

**DOI:** 10.3390/ma16103760

**Published:** 2023-05-16

**Authors:** Ahmed Hussain Jawhari, Nazim Hasan

**Affiliations:** Department of Chemistry, Faculty of Science, Jazan University, Jazan 45142, Saudi Arabia; ahjawhari@jazanu.edu.sa

**Keywords:** hydrogen production, metal electrocatalysts, hydrogen evolution reactions, nanocomposites

## Abstract

Hydrogen is considered a good clean and renewable energy substitute for fossil fuels. The major obstacle facing hydrogen energy is its efficacy in meeting its commercial-scale demand. One of the most promising pathways for efficient hydrogen production is through water-splitting electrolysis. This requires the development of active, stable, and low-cost catalysts or electrocatalysts to achieve optimized electrocatalytic hydrogen production from water splitting. The objective of this review is to survey the activity, stability, and efficiency of various electrocatalysts involved in water splitting. The status quo of noble-metal- and non-noble-metal-based nano-electrocatalysts has been specifically discussed. Various composites and nanocomposite electrocatalysts that have significantly impacted electrocatalytic HERs have been discussed. New strategies and insights in exploring nanocomposite-based electrocatalysts and utilizing other new age nanomaterial options that will profoundly enhance the electrocatalytic activity and stability of HERs have been highlighted. Recommendations on future directions and deliberations for extrapolating information have been projected.

## 1. Introduction

Sustainable human growth is centered around energy and the environment [1]. Oil, coal, and natural gas still supply over 85% of the world’s energy. Chemical industries use fossil fuels. These non-renewable fuels may deplete quickly if burned. Fossil fuel combustion releases greenhouse gases such as CO_2_, NO_2_, and SO_x_, which harm the environment. Renewable carbon-neutral energy should replace fossil energy [2]. Hydrogen energy is a possible replacement for fossil fuels due to its advantages, such as abundance, high gravimetric energy density, and carbon-free emission [3,4]. In addition, 95% hydrogen gas is procured from fossil fuels, which causes serious environmental issues. Water electrolysis using renewable energy sources such as sun or wind can produce ecofriendly hydrogen for the hydrogen economy. Only solar energy can replace fossil fuels. Sunlight is diffusive, which is why energy capture and storage are considered important [3].

Photovoltaic (PV) cells convert solar energy into electricity; however, expensive energy storage devices such as batteries are requited for storage and distribution purposes. As an alternative is this, solar energy is converted to chemical energy, such as H_2_. H_2_ is easy to store and transport, has a high energy density (approx. 140 MJ/Kg at 700 atm), and emits no carbon (the only combustion product of H_2_ is water) [4]. H_2_ is both a chemical feedstock and a fuel for hydrogen fuel cells [5]. PV-based electricity-driven water splitting is promising for H_2_ production, especially because PV electricity is cheaper (0.15 USD kWh@1 (data from LONGi Silicon Materials Corp., Xi’an, China)). Renewable H_2_ production may be best achieved by coupling solar PV electricity with a water electrolyzer (PV-E) [6].

Current water electrolysis technology includes: (1) proton exchange membrane (PEM) electrolysis; (2) alkaline electrolysis; and (3) high-temperature solid oxide water electrolysis. PEM-based electrolysis cells split water in acidic conditions. Lower gas permeability and strong proton conductivity are advantages of this state. It produces hydrogen quickly and efficiently [7]. OER electrocatalysts in acidic media are noble metal and noble metal oxide catalysts. This need makes the cell expensive [8]. Alkaline electrolysis cells divide water. Alkaline water splitting allows non-noble metals or metal oxides to be used as electrocatalysts. Alkaline media usually have 2–3 orders of magnitude less HER activity than acidic media [9]. Thus, designing low-cost, high-catalytic-activity, and durable electrocatalysts for water splitting (electrocatalytic) in diverse mediums is difficult but has always been attempted [10,11,12,13]. Figure 1 represents the schematic of the photocatalytic water splitting method for H_2_ production.

Water electrolysis has two half-cell reactions: a hydrogen evolution reaction (HER) and an oxygen evolution reaction (OER). The cathode reduces water to H_2_ and the anode oxidizes it to O_2_. High overpotentials, which measure kinetic energy barriers, slow OER and HER reaction rates, making water splitting impractical. OERs and HERs depend on catalysis. To produce H_2_ and O_2_, efficient catalysts must reduce OER and HER overpotentials.

Nanometal-based composites have impacted various aspects of electrocatalysis; hence, there are huge opportunities and plenty of unexplored aspects. While platinum is the most effective catalyst for HERs, its high cost and its limited abundance hinder the widespread deployment of Pt-based electrolysis devices. It is in this direction that composite/nanocomposite electrocatalysts are being considered alternative options in order to reduce the cost as well as enhance the catalytic properties.

The objective of the following review is to briefly present an overview of the current status of the metal-based electrocatalysts for HERs, and the advantages and disadvantages of these are presented. The options for HER electrocatalysts from noble and non-noble metals have been listed. The composite/nanocomposite options, both metal-based and non-metal-based, have been gathered, consolidated, and presented. The nano-metal options for HER electrocatalyst have also been compiled, and the future scope and challenges are presented.

## 2. Basic Reactions Involved in HERs

Two half-reactions make up electrochemical water splitting (EWS): the cathodic hydrogen evolution reaction (HER) and the anionic oxygen evolution reaction (the HER is a two-electron transfer reaction that involves the adsorption of H_2_O (alkaline solution) or H_2_ (acidic solution) species on the cathode (the Volmer step) and the desorption of H_2_ from the cathode via chemical (the Tafel step) or electrochemical (the Heyrovsky step) processes. Experimental and computational Tafel slope values reveal the HER mechanism. The four-electron transfer OER mechanism is more complex and highly dependent on electrocatalysts. The two most widely known OER mechanisms are lattice oxygen involvement and adsorbate development. The lattice oxygen involvement process hypothesizes that the oxygen partly comes from the lattice oxygen in metal oxides, while the adsorbate evolution mechanism suggests that the water in electrolytes generates oxygen molecules [14]. H*, O*, HO*, and HOO* are important reaction intermediates in HERs and OERs.

The two-electron transfer half-reaction HER produces hydrogen at the cathode in water electrolysis. The environment strongly affects this HER’s mechanism. Three HER reaction stages are feasible in acidic media.
H^+^ + e^−^ = H_ad_, H^+^ + e^−^ = H_ad_,(1)
H^+^ + e^−^ + H_ad_ = H_2_, H^+^ + e^−^ + H_ad_ = H_2_,(2)
2H_ad_ = H_2_. 2H_ad_ = H_2_.(3)

The first Volmer step (1) produces adsorbed hydrogen. Then, the HER is preceded by the Heyrovsky step (2) or the Tafel step (3), or both, to produce H_2_. In alkaline media, two possible reaction steps, i.e., the Volmer step (4) and the Heyrovsky step (5), are possible, as shown in the following equations, respectively:H_2_O + e^−^ = OH^−^ + H_ad_, H_2_O + e^−^ = OH^−^ + H_ad_,(4)
H_2_O + e^−^ + H_ad_ = OH^−^ + H_2_. H_2_O + e^−^ + H_ad_ = OH^−^ + H_2_.(5)

Theoretical simulations linked HER activity to hydrogen adsorption (H_ad_). Hydrogen evolution materials are usually described by their adsorption of free hydrogen energy (ΔGH). The HER benefits from moderate hydrogen binding energy. Pt is the best HER catalyst in both media with excellent hydrogen adsorption energy and a high exchange current density. Alkaline media had lower HER activity than acidic media [9]. The slow water dissociation phase slows the process by 2–3 orders of magnitude. Industrial plants prefer alkaline electrolysis. Figure 2 presents an overview of the basic HER.

## 3. Electrocatalysts in HERs

HER electrocatalysts fall within two categories, either noble- or non-noble-metal-based catalysts. Several techniques are being researched to improve the HER and lower the price of noble metal electrocatalysts, especially Pt-based catalysts. For instance, alloying Pt with low-cost transition metals could boost Pt consumption and change electrical surroundings, thus improving activity. Coupling Pt with other water dissociation promoters improves alkaline HER activities, which is useful in industries. Due to their low cost and availability, non-noble metal HER electrocatalysts have received a lot of attention [15].

### 3.1. Noble Metal Electrocatalysts

Platinum group metals (PGMs), including Pt, Pd, Ru, Ir, and Rh, hold a reputation as excellent HER catalyzers. Pt tops the volcanic curve. These noble-metal-based catalysts are expensive and hard to store, limiting their commercial use. Alloying Pt with transition metals improves Pt usage changes the electronic microenvironment to boost HER electroactivity. Several variations/combinations of PGMs have been tested for electrocatalyst applications. Sun et al. demonstrated the in situ development of an ultrafine-PtNi-nanoparticle-decorated Ni nanosheet array on a carbon cloth (PtNi-Ni NA/CC) with ultralow loading Pt (7.7%) and improved HER activity (38 mV in 0.1 M KOH at 10 mA cm^−2^) compared to Pt/C (20%), with long-term durability. The downshift of Pt’s D-band center decreases the adsorption energy of oxygenated species (OH*) on the surface Pt atom, which promotes HER performance [16].

For Pt electrocatalysts, HER activity in alkaline media is usually lower than in acidic media [17]. Inefficient water dissociation on the Pt surface reduces HER activity. To increase alkaline HER activity, Pt is usually coupled with water dissociation promoters [15]. Controlling the metal composition on the surface of Pt-based HER electrocatalysts [18,19,20] improves electrocatalytic activity. Subbharaman et al. [21] produced nano Ni(OH)_2_ clusters on Pt electrode surfaces and showed a factor of 8 increase in HER activity compared to Pt. Ni(OH)_2_ cluster edges dissociate water to generate M–H^ad^ intermediates on Pt, and the adsorbed hydrogen intermediates produce H_2_. Zhao et al. created surface-engineered PtNi–O nanoparticles with an enhanced NiO/PtNi interface via Ni(OH)_2_ and Pt(111) synergy. In alkaline environments, this interface structure becomes Ni(OH)_2_, forming a Pt(111)-like interface on the surface. The catalyst has a low HER overpotential of 39.8 mV at 10 mA cm^−2^ with 5.1 µgPt cm^−2^ loading of Pt [22]. Doping Pt-based compounds with other metals can also boost HER catalytic performance, minimizing Pt consumption. N-modified PtNi nanowire catalysts increase water dissociation kinetics via N-induced orbital tuning, revealing an ultralow overpotential of 13 mV at 10 mA cm^−2^ in alkaline environments [14].

### 3.2. Non-Noble Metal Electrocatalysts

Transition metal carbides (TMCs) are widely used to develop non-noble metal electrocatalysts. Mo_2_C and WC exhibit significant HER catalytic activity. Besides strong electrical conductivity, their hydrogen adsorption and d-band electronic density state (similar to Pt) are optimum, which is thought to be a fundamental reason for their high HER activity [23]. Carbon atoms in lattice interstitials provide them with D-band electronic density states akin to the Pt benchmark [24]. Vrubel and Hu (2012) [25] found good HER catalytic activity in acidic and alkaline conditions in commercial molybdenum carbide microparticles (com-Mo_2_C) with a cathodic current of 10 mA cm^−2^ and an overpotential of 90–230 mV. In an attempt to expose more active areas to optimize Mo_2_C catalysts, materials were nanoengineered. Xiao et al. [26] used the hydrogen carburization of anilinium molybdate to make porous Mo_2_C nanorods; commercial Mo_2_C is less active than nanorods. After 2000 cycles, the activity is steady, indicating a good cycling life. In a 1M KOH alkaline environment, the Mo_2_C nanorod outperforms commercial Mo_2_C. High-conductivity and well-defined porous shapes make Mo_2_C nanorods competitive for HERs in acidic and alkaline conditions. Ni nanoparticles could also boost catalytic activity. The HER could be improved as a hybrid nano-electrocatalyst by depositing molybdenum carbides on carbon-based materials. Liu et al. [27] grew molybdenum carbide (Mo_2_C) nanoparticles in situ on graphene nanoribbons (GNRs) via hydrothermal synthesis and high-temperature calcination. In acidic, basic, and neutral conditions, the Mo_2_C-GNR combination was highly electrocatalytic and durable. GNRs as templates for in situ carbide growth are intriguing because the interconnected GNR network structure forms the platform for multiple conductive pathways for fast electron transportation with an extensive active surface area with increased active sites, enhancing catalytic activity in all acidic, basic, and neutral media.

Transition metal phosphides (TMPs) make up one of the fastest-growing fields for producing electrocatalysts with excellent catalytic activity and stability in acid and base solutions (pH-universal). Due to its high conductivity and unusual electrical structure, P atoms are of importance with respect to TMP. Ni_2_P catalysts for the HER were first discovered in 2005. Liu and Rodriguez used a density functional theory (DFT) to study electrocatalysts [28]; their results confirmed that [NiFe] hydrogenase exhibited the highest activity toward the HER, followed by [Ni(PNP)_2_]^2+^ > Ni_2_P > [Ni(PS^3^*)(CO)]^1−^ > Pt > Ni in a decreasing sequence. Strong bonding increases hydrogen desorption energy from metal surfaces. Popczun et al. demonstrated that Ni_2_P nanoparticles placed onto a Ti foil substrate with good HER activity had an exchange density of 3.3 × 10^−5^ mA cm^−2^ and a Tafel slope of 46 m V dec^−1^ [29]. In alkaline electrolytes, the Ni_2_P/Ti electrode’s stability was poor. Hu et al. [30] developed a bimetallic-structured phosphide electrocatalyst, a NiCo_2_P_x_ pH-universal HER catalyst, which was durable and stable in various electrolytes. In acidic, alkaline, and neutral media, NiCo_2_P_x_ performed well. In the alkaline electrolyte, NiCo_2_P_x_ had the lowest overpotential of 58 mV at 10 mA cm^−2^ compared to CoP_x_ (94 mV), NiP_x_ (180 mV), and commercial Pt (70 mV). After 5000 cycles under diverse conditions, NiCo_2_P_x_’s catalyst structure remained intact. The dangling P atom (P^−^) and H atom interact, similar to the under-coordinated metal center (M+, M = Ni, Co) and the O atom. Water molecules dissociate into H atoms and OH^−^, weakening the H–OH bond. Adsorbed H atoms unite to create molecular H_2_ when the H atom is moved to a neighboring unoccupied metal site [14].

Transition metal chalcogenides (sulphides and selenides) are also alternative options. MoS_2_ is a promising HER electrocatalyst and has free energy similar to Pt. Chorkendorff et al. [31] produced triangular MoS_2_ single crystals of varied sizes on Au(111) substrates to locate the structure’s active site. The MoS_2_ catalyst’s edge locations linearly affect electrocatalytic HER activity. Xie et al. [32] identified flaws in MoS_2_ ultrathin nanosheets to boost their electrocatalytic HER performance. The defect-rich structure’s active edge sites were created by the partial breaking of the catalytically inert plane. In another study, Jin et al. [33] produced CoS_2_ with tunable film, microwire (MW), and nanowire (NW) morphologies. They extensively analyzed their structures, activities, and stabilities, and found that the unique morphologies enhance activity and stability. CoS_2_ NWs have the best HER catalytic performance and stability because of their highly effective electrode surface area. HER activity can be increased via heteroatom doping. Oxygen atom inclusion and controlled disorder engineering can govern the electrical structure of MoS_2_ ultrathin nanosheets, increasing conductivity and HER activity [34]. The disordered structure provides huge amounts of unsaturated sulfur atoms as active sites for HERs and a quasiperiodic nanodomain layout for rapid interdomain electron transport. Figure 3 gives an overview of the various metals and non-metallic catalyst supports used thus far for HER applications.

## 4. Composite HER Electrocatalysts

Due to their activity, quantity, and accessibility, nonprecious metals have helped develop new energy materials [35]. Single-metal-atom materials have excellent atom utilization efficiency, high activity, and well-defined active sites, making them promising catalysts [36,37]. Suitable supports (e.g., metal sulfides, hydroxides, g-C_3_N_4_, etc.) are needed to disseminate and stabilize metal active sites with high surface free energy [36,38]. Many single-atom catalysts (SACs) have shown significant catalytic activity and structural stability [39,40]. Qi et al. [41] presented a covalently bonded atomic cobalt array interface catalyst via a phase change in MoS_2_ to metallic D-1T from semiconductive 2H. Yi et al. [42] observed that cobalt SACs with CoeN_4_ moiety had good HER performance, with a 21 mV onset overpotential (h0) and a 50 mV decade1 Tafel slope. These examples demonstrate the potential of SACs towards large-scale water electrolyzers.

Du et al. [43] co-electrodeposited nanoparticles of Ni, HG, and rGO layers on Ni foam to create Ni-HG-GO-Ni foam catalysts. The catalysts performed well in alkaline solution due to their datura-like structure and excellent charge transfer between Ni and HG-rGO. Besides placing active components on conductive supports, heteroatom doping boosts TM catalytic activity. Jin et al. [44] optimized the Ni metal HER activity using a multidimensional heteroatom-doping technique and found that N,P-co-doped Ni NPs performed best (h10 = 24 mV). Theoretical studies revealed that the doping-induced redistribution of charges on the Ni surface results in upgraded characteristics. Alloying is a smart technique that is used to boost TM catalytic activity and lifetime by combining electroactive metals or advancing the ratio of real-to-geometric surface areas [45]. Hundreds of trace metal alloys, especially Fe-, Ni-, and Co-based alloys, perform well for EWS. Hsieh et al.’s cobalt SACs with CoeN_4_ [46] had strong HER and OER activities in basic media and a current density of 10 mA cm^2^ for water splitting at 1.47 V, outperforming the Pt/C and IrO_2_ pair.

Gao et al. [47] designed a modular electrocatalyst with OER/HER catalytic activity by rationally targeting various metal oxide components. The bifunctional CoeCueW oxide catalyst that could bring about water splitting in alkaline water electrolyzers had low overpotential, good faradaic efficiencies, and long stability. Metal-layered double hydroxides (LDHs), such as CoSe@NiFe LDH nanoarrays [48] and Ni@NiFe LDH [49], follow these design ideas. Metal chalcogenides (e.g., sulphides, selenides, and telluride), are more promising as HER catalysts than the corresponding metal oxides/hydroxides [50]. Electrodeposition–annealing–nitridation produced NiCo-nitride/NiCo_2_O_4_-supported graphite fibers; this three-component system supported electroactive sites, such as NiCo_2_O_4_, CoN, and Ni_3_N, at the interfaces between components. The 3D oxide nitride/graphite fibers were established as pH-universal bifunctional electrocatalysts for water splitting [51]. EWS-performing hybrids include Co4N@nitrogen-doped carbon [52] and CoP/Ti_3_C_2_ MXene [53].

### 4.1. Nanocomposites for HER Applications

#### 4.1.1. Pt—Composites

Zhao et al. annealed PtNi/C structures to create octahedral nanomaterial with Ni(OH)_2_-Pt(111)-like interfaces. PtNi-O/C material has a mass activity of 7.23 mA/μg at 70 mV, which is almost eight times greater than that of commercial Pt/C catalysts [22]. Zhang et al. showed how regulated synthesis could affect a catalyst’s electrocatalytic activity and stability towards the HER under alkaline circumstances. They synthesized well-crystalline lotus–thalamus-shaped Pt-Ni anisotropic superstructures using a solvothermal technique and recorded an overpotential of 27.7 mV at 10 mA cm^−2^ and a turnover frequency of 18.63 H_2_ s^−1^ at 50 mV [54]. Koo et al. used alkyltrimethyl-ammonium bromide, K_2_PtCl_4_, and NaBH_4_ to make different-sized Pt nanocubes. The crop-casting of 8, 20, and 25 nm Pt nanocubes was carried out on a FTO/glass substrate. Under optimum conditions, intermediate-size nanocubes, surrounded by {100} facets with some corners chopped to expose {110} facets, produced 1.77 A mg^−1^ at 50 mV and 0.54 A mg^−1^ at 100 mV. Nanocubes and cuboctahedra were observed [55], and catalytic performance was found to depend on shape effect, and, as reported in catalysis, size is crucial [56,57].

Zhang et al. used electrochemical deposition to immobilize Pt SAs over two-dimensional inorganic material (MXene-Mo_2_TiC_2_Tx) nanosheets using Mo vacancies. The catalyst’s high HER catalytic activity, with modest overpotentials of 30 and 77 mV to achieve 10 and 100 mA cm^−2^, was attributable to strong covalent connections between the Pt SAs and the support. This prevented atoms from aggregating. The material had roughly 40 times greater mass activity than a Pt/C catalyst [58]. Yu et al. facilitated the chemical binding of 2,6:2′,2″-terpyridine onto a 3D carbon support in a single step, which was submerged in an aqueous K_2_[PtCl_4_] solution with 0.5 ppm Pt^2+^ for 2 h at room temperature to create Pt SAs. They achieved a very low metal loading (0.26 ± 0.02 μg·cm^−2^ of Pt) with a mass activity of 77.1 A·mgPt^−1^ at 50 mV [59]. Elmas et al. synthesized a platinum-group metal-selective electropolymerizable monomer (4-(terthiophenyl)-terpyridin) with a pendant terpyridine unit by utilizing a metal-selective ligand approach. The metallopolymer had highly active Pt SAs that catalyzed the HER [60].

Cheng et al. used ab atomic layer deposition to produce Pt SAs and clusters on nitrogen-doped graphene nanosheets for the HER and found that almost all the Pt atoms were used, which was economically appealing and 37 times more active than commercial Pt/C catalyst [61]. Sun et al. used atomic layer deposition to immobilize Pt SA on graphene nanosheets (undoped). Their catalytic activity was 10 times higher than that of the commercial Pt/C catalyst [62]. According to Shi et al., site-specific electrodeposition on two-dimensional transition metal dichalcogenides supports yielded Pt SAs (MoS_2_, WS_2_, MoSe_2_, and WSe_2_). By tuning the metal’s d-orbital state, the anchoring chalcogens (S and Se) and transition metals (Mo and W) can synergistically regulate Pt SAs’ electronic structure. Pt SAs outperformed MoSe_2_ with 34.4 A mg^−1^ under an overpotential of 100 mV [63]. Despite significant breakthroughs, wet chemical synthesis is preferred because of its simplicity [39,64].

#### 4.1.2. Palladium-Based Composites

Pd’s atomic size is similar to Pt’s, making it a good HER catalyst. Pd has various efficient synthetic methods for size- and shape-controlled nanocrystal creation [65,66,67,68]. Pd nanoctahedra bordered by (111) facets have a high H loading of 0.90, making them attractive for HER applications [69]. Li et al. created a core@shell PdCu@Pd nanocube catalyst for efficient HERs. The system needed an overpotential of 10 mV to reach 68 mA cm^−2^ [70]. Wang et al. used a polyol technique with polyvinylpyrrolidone as a stabilizer to make Pd icosahedral NPs with 0.22 nm lattice spacing in the (111) plane. After 130,000 cyclic voltammetric cycles, the overpotential was 32 mV at 10 mA cm^−2^, confirming its durability and activity [71].

#### 4.1.3. Nickel-Based Composites

Due to its chemical characteristics and group numbers being similar to Pt, Ni is a potential non-noble metal choice for HERs. It is cheaper and more abundant than Pt and Pd. In alkaline circumstances, the rate-determining step is hydrogen adsorption (the Volmer step), and the low-valence-state oxide of Ni improves the HER [72]. The reaction is interesting for Ni–Pt and Ni–Mo alloys, as well as Ni-nitride-, Ni-oxide-, Ni-phosphide, and Ni-sulfide-based catalysts [73]. Li et al. published an intriguing study based on the Cu–Ni alloy. Selective wet chemical etching produced edge-notched, edge-cut, and mesoporous Cu–Ni nanocages. The highly catalytic (111) facets were etched from the last two materials. The Cu–Ni nanocages had increased HER activity under alkaline circumstances (a current density of 10 mA cm^−2^ under an overpotential of 140 mV). The density functional theory showed that Ni–Cu alloys are more promising than pure Ni [74]. Nanocubes of Ni(OH)_2_ and Ni–Fe modified with Pt atoms were proposed for the process to reduce noble metal loading [75,76]. After alloying with Pt, Kavian et al. generated 9 nm Pt–Ni octahedral nanocrystals with 15-fold higher specific activity and roughly 5 times more mass activity in alkaline media than the Pt/C commercial catalyst [77]. Considering the overpotential, Tafel slope, and turnover frequency, Seo et al. found that spherical nickel phosphide nanocrystals with (001) facets had higher HER activity than rod-shaped ones with (210) facets [78]. Xiang et al. produced nickel phosphide nanowires using a one-step hydrothermal method with an overpotential of 320 mV and a Tafel slope of 73 mV dec^−1^ for HERs [79,80]. Table 1 summarizes a comprehensive list of the various electrocatalysts that have been used for HER applications.

**Table 1 materials-16-03760-t001:** The HER performance of predominant electrocatalysts.

Catalysts	HER Performance	Stability	Reference
PtNi-Ni NA/CC	10 I mA cm^−2^; 38η mV; Tafel slope of 42 mV dec^−1^	90 h	[16]
PtNi-O/C	10 I mA cm^−2^; 39.8η mV; Tafel slope of 78.8 mV dec^−1^	10 h	[22]
PtNi(N) NW	10 I mA cm^−2^; 13η mV; Tafel slope of 29 mV dec^−1^	10 h	[15]
Mo_2_C-R	32 I mA cm^−2^; 200η mV; Tafel slope of 58 mV dec^−1^	2000 cycle	[25]
Mo_2_C-GNR	10 I mA cm^−2^; 266η mV; Tafel slope of 74 mV dec^−1^	3000 cycles	[26]
Ni_2_P/Ti	20 I mA cm^−2^; 130η mV; Tafel slope of 46 mV dec^−1^	500 cycles	[29]
NiCo_2_Px	10 I mA cm^−2^; 63η mV; Tafel slope of 63.6 mV dec^−1^	5000 cycles	[30]
Defect rich MoS_2_	13 I mA cm^−2^; 200η mV; Tafel slope of 50 mV dec^−1^	10,000 s	[32]
CoS_2_ NW	10 I mA cm^−2^; 145η mV; Tafel slope of 51.6 mV dec^−1^	3 h	[33]
CoS_2_ NW	10 I mA cm^−2^; 158η mV; Tafel slope of 58 mV dec^−1^	41 h	[33]
Oxygenated MoS_2_	120η mV; Tafel slope of 55 mV dec^−1^	3000 cycles	[34]
Pt SAs over nanosheets of a two-dimensional inorganic material MXene-Mo_2_TiC_2_T_x_	Overpotential of 77 mV is 8.3 Amg^−1^, 39.5 times more than commercial HER catalyst (40 wt% Pt/C, 0.21 Amg^−1^); Tafel slope of 30 mVdec^−1^	10,000 cycles	[58]
PdCu@Pd nanocube core@shell	10 I mA cm^−2^; 65η mV; Tafel slope of 35 mV dec^−1^	ND	[70]
Polyvinylpyrrolidone as a stabiliser to make Pd icosahedral NPs	32 mV at 10 mA cm^−2^	130,000 cycles	[71]
Ni_2_P NW	320 mV and a Tafel slope of 73 mV dec^−1^	ND	[79]
Cu-Ni nanocages	Current density of 10 mA cm^−2^ (under an overpotential of 140 mV).	ND	[74]
Pt SA on graphene nanosheets	10 times more active than commercial Pt/C	150 cycles	[62]
Pt SAs and clusters on nitrogen-doped graphene nanosheets	37 times more active than Pt/C	50 and 100 cycles	[61]
Pt nanocubes on FTO/glass substrate	1.77 A mg^−1^ at 50 mV and 0.54 A mg^−1^ at 100 mV	ND	[55]
Ni-HG-rGO/NF catalysts	−10 and −100 mA cm^−2^; overpotentials of −50 and −132 mV; a low Tafel slope of −48 mV dec^−1^	ND	[43]
Cobalt SACs with a Co-N_4_	21 mV onset overpotential (h_0_) and a Tafel slope of 50 mV dec^−1^	10 h	[42]
NiFeMo alloy	Ultralow overpotentials of 33 and 249 mV; 500 mA cm^−2^	50 h	[46]
NiCo-nitrides/NiCo_2_O_4_/GF	Low Tafel slope of 58 mV dec^−1^; lowest overpotentials (η) of 71 and 180 mV to obtain current densities of 10 and 50 mA cm^−2^	over 40 h	[51]
Ni@Pd/PEI–rGO stack structures	10 I mA cm^−2^; 90η mV; Tafel slope of 54 mV dec^−1^	ND	[81]
MWCNTs@Cu@MoS_2_	10 I mA cm^−2^; 184η mV; Tafel slope of 62 mV dec^−1^	1000 cycles	[82]
Nanoporous Ag_2_S/CuS	10 I mA cm^−2^; 200η mV; Tafel slope of 75 mV dec^−1^	1000 cycles	[83]
Wl_8_O_49_@WS_2_ NRs	10 I mA cm^−2^; 310η mV; Tafel slope of 86 mV dec^−1^	750 cycles	[84]
RuCo/Ti foil	10 I mA cm^−2^; 387η mV; Tafel slope of 107 mV dec^−1^	>12 h	[85]
Rh_2_S_3_–Thick HNP/C	10 I mA cm^−2^; 122η mV; Tafel slope of 44 mV dec^−1^	10,000 cycles	[86]
Fe_1_-xCoxS_2_/CNT	10 I mA cm^−2^; 158η mV; Tafel slope of 46 mV dec^−1^	>40 h	[87]
MoS_2_@OMC	10 I mA cm^−2^; 182η mV; Tafel slope of 60 mV dec^−1^	ND	[88]
Pd_2_Te NWs/rGO	10 I mA cm^−2^; 48η mV; Tafel slope of 63 mV dec^−1^	48 h	[89]
NiAu@Au NPs	Tafel slope of 36 mV dec^−1^	20,000 cycles	[90]

ND—not determined.

## 5. Future Perspectives and Conclusions

Developing advanced low-cost electrocatalysts for water splitting is of great scientific and industrial significance. The existing metal composites/nanocomposite electrocatalysts used for HER applications were reviewed and discussed in this review. The main concern is not a dearth in options; it instead rests on criteria such as cost-effective and environmentally friendly electrocatalyst options. The realization of the successful commercialization of low-cost electrocatalysts deserves attention. First, developing simple and scalable synthetic methods that can enable mass production is mandatory when it needs to scale up for large-scale applications. Secondly, in-depth knowledge on the strengths and weaknesses of electrocatalyst options is required. Thirdly, catalyst stability with respect to industrial protocols (e.g., time, temperature, mass loading) is another crucial mandate. Finally, developing bifunctional/multifunctional catalysts in view of simplification, cost reduction, and practical application is also necessary.

Metal-free catalysts are potential advanced energy sources due to their abundance, cost-effectiveness, pH resistance, and environmental friendliness [91,92,93]. Carbon materials, such as multiwalled/single-walled CNTs, graphene, carbon quantum dots, graphene oxide (rGO), and graphdiyne, are used to make advanced electrocatalysts due to their excellent electrical and thermal conductivity, low cost, large active surface area, good mechanical/chemical strength, and tunable electronic structures [92,94,95,96,97,98]. Heteroatom (N, O, S, P, B, or F) doping and defect structure fabrication are the most effective ways to tune the electrical structure of bare carbon [99,100]. P doping increased graphite layer HER activity [101]. Due to its high conductivity, configurable direct bandgap, and anisotropic characteristics, phosphorene is becoming an electrocatalyst alternative to carbon materials [80]. Phosphorene-based catalysts exhibit good HER and OER performance. Doping and defect engineering can boost phosphorene activity, like carbon [80,99,100,101]. Through the course of the review, we observed that most of the studies with respect to material options for HER electrocatalysts report one material after the other in various combinations, with varied dopants and defects; there is no systematic study at present that compares the efficacy of these metal electrocatalysts and puts forth optimized catalyst suggestions. This is ideally important when it comes to the strategic planning of next-generation catalysts. Moreover, the catalytic mechanisms of many electrocatalysts, such as metal-based catalysts for HERs in alkaline conditions, have been hardly researched compared to HERs in acidic conditions. This is another aspect that deserves more research attention.

Nanotechnology has transformed every sector, but catalysis has benefited the most [80,102]. Metal NPs seek customizable characteristics with controllable parameters for better catalysis. A lack of theoretical models of electronic structures, the controlled synthesis of the shape and presentation of specific facets, and a fundamental grasp of catalytic processes hinder catalyst development. Electrocatalyst nanomaterials produce surface species with suitable bonding energies. Transition metals (TMs), such as Fe-, Ni-, and Co-based and metal-free catalysts, electrocatalyze HERs and/or OERs, while metal-free catalysts (e.g., N, P, S, and O) incorporate carbon nanomaterials, and other options come from TMs, TM alloys, and TMX (X = O, S, Se, Te, N, P, B, and C). Several of these low-cost chemicals have EWS capabilities comparable to NM-based catalysts. We have reported the effective use of Pt-Ag/Ag_3_PO_4_-WO_3_ nanocomposites for photocatalytic H_2_ production from bioethanol [103]. Such metal-based nanocomposites hold huge relevance and similar such trinary composites will combine multifold properties of materials involved, and hence will be able to contribute significantly to this application. Different heterostructures based on WO_3_ with noble metals such as Pt/TiO_2_/WO_3_ [104], TiO_2_@WO_3_/Au [105], Ag_3_PO_4_/Ag/WO_3−x_ [106], and Ag/Ag_3_PO_4_/WO_3_ [107] were reported to exhibit excellent photocatalytic potential for the photodegradation of organic compounds. Such nanocomposites should be extended to electrocatalyst applications. We recently published a new study [108], titled Noble Metals Deposited LaMnO_3_ Nanocomposites for Photocatalytic H_2_ Production. Similar lanthanum-based nanocomposites need to be tested for HER electrocatalyst applications too.

Previous authors report that similar morphologies of the same materials yielded different tendencies. So, the question based on what is behind such findings remains. This will require a proper understanding of the interactions between the surface facets/morphologies of supports and active species. In situ studies are critical in order to obtain an advanced comprehension of catalytic processes and to improve the catalysts’ design and performance. Strategies that aim to regulate the internal and external characteristics of nanomaterials are recommended, such as heteroatom doping, hybridization, defect engineering, phase control, and nanostructure construction.

Our PubMed search using various keywords (metal catalysts in a hydrogen evolution reaction, composite catalysts in a hydrogen evolution reaction, and nanocomposite catalysts in a hydrogen evolution reaction) provided varying amounts of search results (2643, 760, and 127, respectively) (Figure 4). This trend reflects the nature of research interests in the corresponding areas. This survey clearly indicates that the use of nanocomposites is significantly lagging. When nanotechnology becomes very promising in various aspects, its use in this application decreases, which is a strange mismatch. Composites largely benefit from their ability to combine unique properties of various materials. Metallic electrocatalysts on non-metallic supports (CNTs and graphene) have been reported to enhance HER applications. Integrating nano-concepts to composites is certainly a fulfilling area, which we found has not been appropriately attempted. This certainly represents a gap in the current setting. With the ever-expanding list of nanomaterials at our disposal, there will be a huge waste of resources if we do not avail them. This review prompts the incorporation of more nano-aspects into electrocatalysts and the improvisation of existing composite catalysts with nanoforms. There is plenty of room when there is nano and superior attributes to harness and benefit sustainable energy production.

## Figures and Tables

**Figure 1 materials-16-03760-f001:**
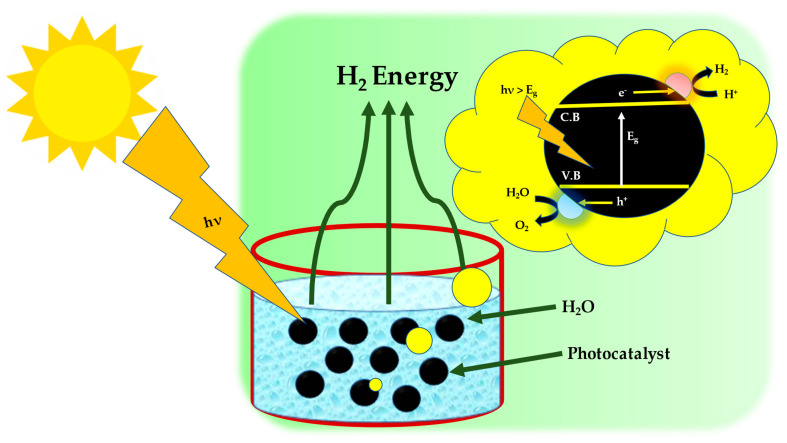
Hydrogen energy obtained by photocatalytic water splitting.

**Figure 2 materials-16-03760-f002:**
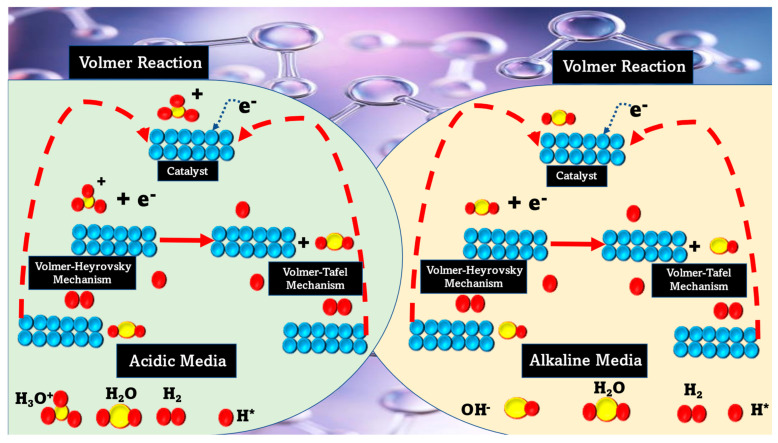
The basic scheme of reaction involved in HERs in alkaline and acidic environments.

**Figure 3 materials-16-03760-f003:**
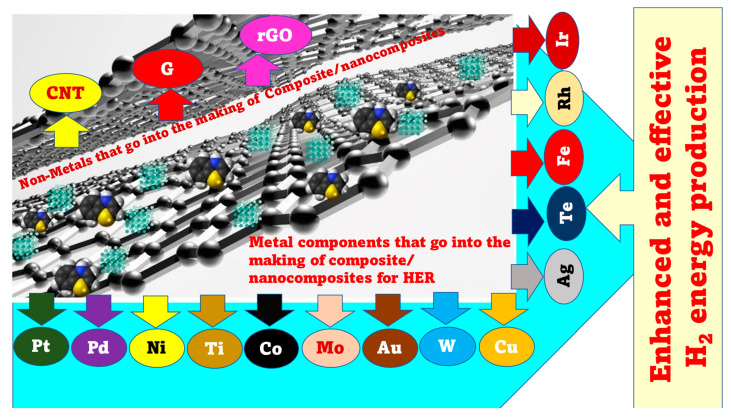
An overview of the metals and non-metals that are used to make composites/nanocomposites for HER applications.

**Figure 4 materials-16-03760-f004:**
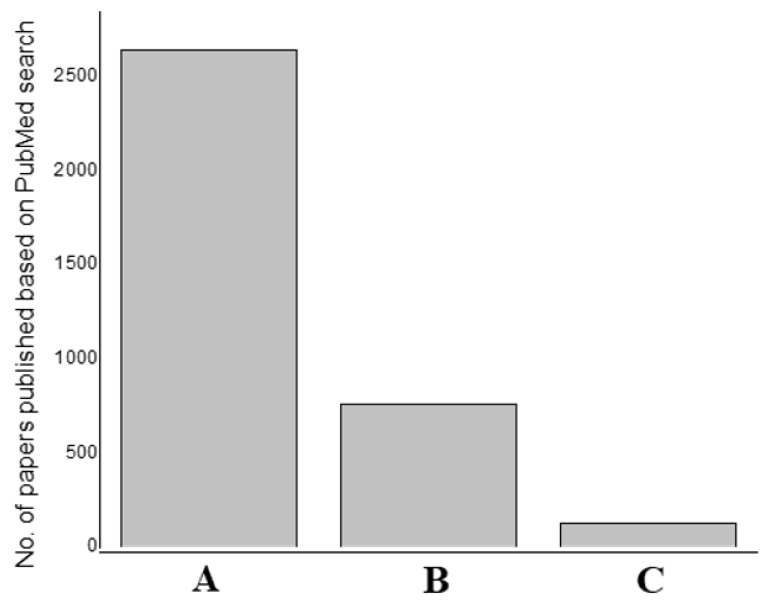
Graph showing the current research trends with respect to composites and nanocomposites for HER applications. A represents PubMed search term results on catalysts in hydrogen evolution reaction; B represents search term results on composite catalysts in hydrogen evolution reaction, and C represents search term results on nanocomposite catalysts in the hydrogen evolution reaction. The values are based on PubMed search (as of April 2023).

## Data Availability

Not applicable.

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
