# Peer review of "Nanocomposite Electrocatalysts for Hydrogen Evolution Reactions (HERs) for Sustainable and Efficient Hydrogen Energy—Future Prospects"

_materials, 2023, doi:10.3390/ma16103760_

Round 1
Reviewer 1 Report
In the manuscript " Nanocomposite metal-based electrocatalysts for hydrogen evolution reaction (HER) for sustainable and efficient hydrogen energy –future prospects" Hasan et al. investigates the development of active, stable, and low-cost catalysts or electrocatalysts to achieve optimized electrocatalytic hydrogen production from water splitting. This manuscript is well-organized and carefully written. It can be accepted after minor revision. The comments are presented as follows:
1. The number of references about water splitting is insufficient and needs to be increased. Such as, Feng Yajie,Duan Youyu,Zou Hanjun,Ma Jiangping,Zhou Kai,Zhou Xiaoyuan. Research Status of Single Atom Catalyst in Hydrogen Production by Photocatalytic Water Splitting. Chinese Journal of Rare Metals. 2021,45(5):551-568. Xiaojie Zhao, Ying Chang, Xiaolong He, Huiqi Zhang, Jingchun Jia, Meilin Jia. Understanding ultra-dispersed CeOx modified iridium clusters as bifunction electrocatalyst for high-efficiency water splitting in acid electrolytes. Journal of Rare Earths, 2023, 41, 208-214.
“Hydrogen energy is a possible replacement for fossil fuels due to its abundance, high gravimetric energy density, and carbon-free emission [3,4].”the latest literature about hydrogen energy should be cited, such as Chao Wan, Liu Zhou, Lin Sun, Lixin Xu, Dang-guo Cheng, Fengqiu Chen, Xiaoli Zhan, Yongrong Yang. Boosting visible-light-driven hydrogen evolution from formic acid over AgPd/2D g-C3N4 nanosheets Mott-Schottky photocatalyst, Chemical Engineering Journal, 2020, 396, 125229.
2. The clarity of the Figure 3 needs to be improved, moreover, the number of figures should be added.
3. The conclusion needs to be condensed.
4. What is the main question addressed by the research?
Author Response
In the manuscript " Nanocomposite metal-based electrocatalysts for hydrogen evolution reaction (HER) for sustainable and efficient hydrogen energy –future prospects" Hasan et al. investigates the development of active, stable, and low-cost catalysts or electrocatalysts to achieve optimized electrocatalytic hydrogen production from water splitting. This manuscript is well-organized and carefully written. It can be accepted after minor revision. The comments are presented as follows:
We would like to humbly thank our dear reviewer for the kind comments and appreciative encouraging words. We have carried out the revision as per your suggestion. Thank you.
- The number of references about water splitting is insufficient and needs to be increased. Such as, Feng Yajie,Duan Youyu,Zou Hanjun,Ma Jiangping,Zhou Kai,Zhou Xiaoyuan. Research Status of Single Atom Catalyst in Hydrogen Production by Photocatalytic Water Splitting. Chinese Journal of Rare Metals. 2021,45(5):551-568. Xiaojie Zhao, Ying Chang, Xiaolong He, Huiqi Zhang, Jingchun Jia, Meilin Jia. Understanding ultra-dispersed CeOx modified iridium clusters as bifunction electrocatalyst for high-efficiency water splitting in acid electrolytes. Journal of Rare Earths, 2023, 41, 208-214.
“Hydrogen energy is a possible replacement for fossil fuels due to its abundance, high gravimetric energy density, and carbon-free emission [3,4].”the latest literature about hydrogen energy should be cited, such as Chao Wan, Liu Zhou, Lin Sun, Lixin Xu, Dang-guo Cheng, Fengqiu Chen, Xiaoli Zhan, Yongrong Yang. Boosting visible-light-driven hydrogen evolution from formic acid over AgPd/2D g-C3N4 nanosheets Mott-Schottky photocatalyst, Chemical Engineering Journal, 2020, 396, 125229.
Ans. thank you for the suggestion, we have added these references whereever appropriate. Thank you.
- The clarity of the Figure 3 needs to be improved, moreover, the number of figures should be added. Ans. Improved and one figure has been added
- The conclusion needs to be condensed. Ans. Section 5 is not only conclusions, its the future perspective and future recommendations too, thats why it appears long. We have rephrased the title to improve clarity.
- What is the main question addressed by the research? Ans. The review addresses the various material options that are available for HER and highlights the material options that are promising yet unattempted. Thank you.
Reviewer 2 Report
In this manuscript, Jawhari et al. provided a review on the recent progress and future perspectives toward the development of nanocomposite metal-based electrocatalysts for the hydrogen evolution reaction (HER), a key half-reaction in electrochemical water splitting for green hydrogen production. Overall, the review topic is new and interesting and can appeal to the readership of the journal Materials. However, some technical issues need to be resolved to further improve the quality of the manuscript. Please see below comments for more detail.
1. The authors attempted to review the recent developments of “Nanocomposite metal-based electrocatalysts” for HER. However, the Abstract and the Introduction were not written in a way that highlights how “Nanocomposite metal-based electrocatalyst” is unique and worthwhile to be reviewed. Therefore, these sections are suggested to be revised.
2. The authors aim to review “Nanocomposite metal-based electrocatalysts” for HER. To improve clarity, a clear definition of this type of “Nanocomposite metal-based electrocatalysts” should be provided. The authors actually mentioned a wide range of HER catalyst candidates, among which some may not fall in this group. For example, Table 1 is supposed to summarize HER performance of composite or nanocomposite materials, but some of the catalysts included cannot be classified into composites or nanocomposites, such as NiCo2Px, defect rich MoS2, NiFeMo alloy, etc.
3. To appeal to a broader readership, very recent works on water electrolysis, HER catalysts, and nanocomposite catalysts are suggested to be referenced in the Introduction (e.g., Energy Technology, 2022, 10, 2200573; Materials Reports: Energy, 2021, 1, 100006).
4. Writing needs to be checked for English. Typos need to be corrected. For instance, “electricity his cheaper” in line 42; “watersplitting” in line 309.
5. Table 1, what is “ND” in the column of stability? In addition, Table 1 caption, “Composite/nanocomposite metals” is not appropriate.
6. The focus of this review paper is about HER, however, in the main text (starting from section 3), the OER was mentioned to some extent (such as in line 219, line 222, line 331, etc.), which is believed to be not relevant.
7. It is suggested that the authors include several more figures to aid the discussion of the recent developments of nanocomposite catalysts in HER. Currently, the figures included in this review paper were just prepared by the authors. Some figures from representative or seminal works should be included.
Author Response
In this manuscript, Jawhari et al. provided a review on the recent progress and future perspectives toward the development of nanocomposite metal-based electrocatalysts for the hydrogen evolution reaction (HER), a key half-reaction in electrochemical water splitting for green hydrogen production. Overall, the review topic is new and interesting and can appeal to the readership of the journal Materials. However, some technical issues need to be resolved to further improve the quality of the manuscript. Please see below comments for more detail.
- The authors attempted to review the recent developments of “Nanocomposite metal-based electrocatalysts” for HER. However, the Abstract and the Introduction were not written in a way that highlights how “Nanocomposite metal-based electrocatalyst” is unique and worthwhile to be reviewed. Therefore, these sections are suggested to be revised. Ans. We have worked on the abstract and introduction. thank you.
- The authors aim to review “Nanocomposite metal-based electrocatalysts” for HER. To improve clarity, a clear definition of this type of “Nanocomposite metal-based electrocatalysts” should be provided. The authors actually mentioned a wide range of HER catalyst candidates, among which some may not fall in this group. For example, Table 1 is supposed to summarize HER performance of composite or nanocomposite materials, but some of the catalysts included cannot be classified into composites or nanocomposites, such as NiCo2Px, defect rich MoS2, NiFeMo alloy, etc. Ans. We have revised and thoroughly checked to ensure clarity and retained what needs to be retained in the revision. thank you.
- To appeal to a broader readership, very recent works on water electrolysis, HER catalysts, and nanocomposite catalysts are suggested to be referenced in the Introduction (e.g., Energy Technology, 2022, 10, 2200573; Materials Reports: Energy, 2021, 1, 100006). Ans. Added. thank you.
- Writing needs to be checked for English. Typos need to be corrected. For instance, “electricity his cheaper” in line 42; “watersplitting” in line 309. Ans. Sorry about that, we have now checked and corrected the entire document. Thank you for your kind patience.
- Table 1, what is “ND” in the column of stability? In addition, Table 1 caption, “Composite/nanocomposite metals” is not appropriate. Ans. Abbreviated and title changed.
- The focus of this review paper is about HER, however, in the main text (starting from section 3), the OER was mentioned to some extent (such as in line 219, line 222, line 331, etc.), which is believed to be not relevant. Ans. We wanted to briefly mention OER since its hard to talk about HER not saying anything about OER. Thank you.
- It is suggested that the authors include several more figures to aid the discussion of the recent developments of nanocomposite catalysts in HER. Currently, the figures included in this review paper were just prepared by the authors. Some figures from representative or seminal works should be included. Ans. We have added one figure as per your suggestion, however since including others work will require copyright ( will take time), we have restricted to our own figures. thank you for your kind understanding.
Reviewer 3 Report
Comments from Reviewer
Title: Nanocomposite metal-based electrocatalysts for hydrogen evolution reaction (HER) for sustainable and efficient hydrogen energy –future prospects
The current form's presentation of methods and scientific results is unsatisfactory for publication in the Materials journal. There were numerous typographical errors. - lack of attention to detail, consistent with the standard of a scientific presentation. After reading the manuscript more than once and comparing it with previous studies, I thought it lacked novelty regarding the prepared material or its application in hydrogen evolution reaction. Therefore, I cannot recommend that this paper be published in Materials.
The minor and significant drawbacks to be addressed can be specified as follows:
1. Line 68. ”(AOR) (OER)”???
2. H2O, H2,... subscripts???
3. Fig. 1. H3O+
4. Citation is bad? For example, (i) Markovic et al. [22] - 22. R. Subbaraman, (…)? (ii) Huang et al. created (…) loading [23]. - 23. Z. Zhao, (…) ? (iii) Hu et al. [26] found (…) - 26. H. Vrubel, (…)
5. Fig. 2. Composite ---> composite. Effective ---> effective.
6. Fig. 3. What time range? The information on the x-axis are illegible. Papers ---> papers.
7. I am surprised that the authors did not cite any of their published papers! Are the authors specialists in this subject to write review papers?
8. References. Literature should also be standardized: the size of letters in the titles of journals, initials of names, the size of letters in the titles of articles
9. I rate the work quite low as a review: no diagrams - a huge amount of texts. No results (drawings/plots/experimental results) of other authors are shown. It's very hard to read!
Sincerely,
The reviewer.
Author Response
Title: Nanocomposite metal-based electrocatalysts for hydrogen evolution reaction (HER) for sustainable and efficient hydrogen energy –future prospects
The current form's presentation of methods and scientific results is unsatisfactory for publication in the Materials journal. There were numerous typographical errors. - lack of attention to detail, consistent with the standard of a scientific presentation. After reading the manuscript more than once and comparing it with previous studies, I thought it lacked novelty regarding the prepared material or its application in hydrogen evolution reaction. Therefore, I cannot recommend that this paper be published in Materials.
Ans. We apologize for the typos, we have now thoroughly revised the manuscript. The novelty of the manuscript stands on the ground that we compile the available data and project the various materials and composites used for HER applications. We further address the gaps and the alternative state of the art new age materials that need to be employed for these applications. thank you.
The minor and significant drawbacks to be addressed can be specified as follows:
1. Line 68. ”(AOR) (OER)”???
2. H2O, H2,... subscripts???
3. Fig. 1. H3O+
4. Citation is bad? For example, (i) Markovic et al. [22] - 22. R. Subbaraman, (…)? (ii) Huang et al. created (…) loading [23]. - 23. Z. Zhao, (…) ? (iii) Hu et al. [26] found (…) - 26. H. Vrubel, (…)
5. Fig. 2. Composite ---> composite. Effective ---> effective.
6. Fig. 3. What time range? The information on the x-axis are illegible. Papers ---> papers.
7. I am surprised that the authors did not cite any of their published papers! Are the authors specialists in this subject to write review papers?
8. References. Literature should also be standardized: the size of letters in the titles of journals, initials of names, the size of letters in the titles of articles
9. I rate the work quite low as a review: no diagrams - a huge amount of texts. No results (drawings/plots/experimental results) of other authors are shown. It's very hard to read!
Ans. We have corrected all these minor comments and we have added a new figure and cited few of our papers relevant. thank you.
Reviewer 4 Report
The paper titled “Nanocomposite metal-based electrocatalysts for hydrogen evolution reaction (HER) for sustainable and efficient hydrogen energy –future prospects” overviews the current status of the metal-based electrocatalysts for HER as well as their advantages and disadvantages. The paper is acceptable for publication after appropriate revision. The comments are listed below.
1. The last sentence in the abstract seems to be incomplete.
2. Indexes in chemical formulae should be subscript.
3. Line 42: “because PV electricity his cheaper” -?-> “is cheaper”.
4. Lines 46-47: punctuation is missing.
5. All the abbreviations should be defined when first used. The use of undefined abbreviations in the abstract should be avoided.
6. Line 74: “two most” -> “Two most”
7. Line 127: “with 5.1 μgpt cm−2 Pt loading” -?
8. What are NieHGerGOeNi, HGerGO, CoeN4, and CoeCueW?
9. Line 246: “was tested” -> “were tested”
10. Line 276: “Pd is inexpensive” – really?
11. I would recommend considering one more paper within this review: Catalysts. 2023. V.13. N3. #599. DOI: 10.3390/catal13030599
Author Response
The paper titled “Nanocomposite metal-based electrocatalysts for hydrogen evolution reaction (HER) for sustainable and efficient hydrogen energy –future prospects” overviews the current status of the metal-based electrocatalysts for HER as well as their advantages and disadvantages. The paper is acceptable for publication after appropriate revision. The comments are listed below.
We thank you for your encouraging words and comments. We have incorporated all your suggestions into the revised manuscript. Thank you.
- The last sentence in the abstract seems to be incomplete. Ans. completed
- Indexes in chemical formulae should be subscript. And. It was alright when we submitted , seems to have lost that formatting. now Changed.
- Line 42: “because PV electricity his cheaper” -?-> “is cheaper”. Ans. Corrected
- Lines 46-47: punctuation is missing. Ans. Added
- All the abbreviations should be defined when first used. The use of undefined abbreviations in the abstract should be avoided. Ans. Yes taken care of
- Line 74: “two most” -> “Two most” Ans. Changed
- Line 127: “with 5.1 μgpt cm−2 Pt loading” -? And- Modified
- What are NieHGerGOeNi, HGerGO, CoeN4, and CoeCueW? Ans. Corrected
- Line 246: “was tested” -> “were tested” Ans. Changed
- Line 276: “Pd is inexpensive” – really? Ans. Sorry typo. Corrected now. thank you very much for your valuable suggestions and corrections.
- I would recommend considering one more paper within this review: Catalysts. 2023. V.13. N3. #599. DOI: 10.3390/catal13030599 Ans. Added. Thank you for your time and patience.
Round 2
Reviewer 2 Report
The revised manuscript can be accepted.
Reviewer 3 Report
My comments have been appropriately addressed in the revised manuscript.